# Real-Life Incident Atrial Fibrillation in Outpatients with Coronary Artery Disease

**DOI:** 10.3390/jcm9082367

**Published:** 2020-07-24

**Authors:** Sandro Ninni, Gilles Lemesle, Thibaud Meurice, Olivier Tricot, Nicolas Lamblin, Christophe Bauters

**Affiliations:** 1CHU Lille, Department of Cardiology, University of Lille, F-59000 Lille, France; gilles.lemesle@chru-lille.fr (G.L.); nicolas.lamblin@chru-lille.fr (N.L.); christophe.bauters@chru-lille.fr (C.B.); 2Institut Pasteur de Lille, U1011, F-59000 Lille, France; 3Hôpital Privé Le Bois, 59003 Lille, France; tmeurice@me.com; 4Centre Hospitalier de Dunkerque, 59240 Dunkerque, France; oliviertricot@gmail.com; 5Institut Pasteur de Lille, U1167, F-59000 Lille, France

**Keywords:** coronary artery disease, atrial fibrillation, prognosis, anticoagulation, antiplatelet therapy

## Abstract

**Background:** The risk, correlates, and consequences of incident atrial fibrillation (AF) in patients with chronic coronary artery disease (CAD) are largely unknown. **Methods and results:** We analyzed incident AF during a 3-year follow-up in 5031 CAD outpatients included in the prospective multicenter CARDIONOR registry and with no history of AF at baseline. Incident AF occurred in 266 patients (3-year cumulative incidence: 4.7% (95% confidence interval (CI): 4.1 to 5.3)). Incident AF was diagnosed during cardiology outpatient visits in 177 (66.5%) patients, 87 of whom were asymptomatic. Of note, 46 (17.3%) patients were diagnosed at time of hospitalization for heart failure, and a few patients (*n* = 5) at the time of ischemic stroke. Five variables were independently associated with incident AF: older age (*p* < 0.0001), heart failure (*p* = 0.003), lower left ventricle ejection fraction (*p* = 0.008), history of hypertension (*p* = 0.010), and diabetes mellitus (*p* = 0.033). Anticoagulant therapy was used in 245 (92%) patients and was associated with an antiplatelet drug in half (*n* = 122). Incident AF was a powerful predictor of all-cause (adjusted hazard ratio: 2.04; 95% CI: 1.47 to 2.83; *p* < 0.0001) and cardiovascular mortality (adjusted hazard ratio: 2.88; 95% CI: 1.88 to 4.43; *p* < 0.0001). **Conclusions:** In CAD outpatients, real-life incident AF occurs at a stable rate of 1.6% annually and is frequently diagnosed in asymptomatic patients during cardiology outpatient visits. Anticoagulation is used in most cases, often combined with antiplatelet therapy. Incident AF is associated with increased mortality.

## 1. Introduction

Atrial fibrillation (AF) is commonly observed in patients with coronary artery disease (CAD) [1,2,3]. Thanks to major therapeutic advances in recent decades [4,5], survival of patients with CAD has increased considerably, leaving more opportunity for the development of age-dependent diseases such as AF. The presence of concomitant AF in CAD patients is important in daily practice. Indeed, it may target higher risk patients for both ischemic and bleeding events and critically affect patient management, especially regarding antithrombotic strategies [1,3,6]. Although the risk, correlates, and consequences of incident AF have been extensively studied in the general population [7,8], and to the best of our knowledge, data are lacking in patients with chronic CAD. In addition, how antithrombotic drugs are managed in contemporary practice in such a setting is not known. The level of evidence of guidelines is indeed very low and practices may, therefore, widely differ between physicians. Given the specificities of the CAD population (routine follow-up by cardiologists, background secondary medical prevention therapy including antiplatelet drugs, prognostic implications of concomitant diseases), we sought to investigate these issues.

We analyzed data for 5031 CAD outpatients without prevalent AF included in a prospective registry. Here, we report the incidence, correlates, diagnostic circumstances, management, and prognostic impact of a first episode of AF occurring during the 3-year study follow-up.

## 2. Methods

### 2.1. Study Population

The CARDIONOR study is a multicenter registry that enrolled 10,517 consecutive outpatients with a diagnosis of CAD, AF, and/or heart failure (HF) between January 2013 and May 2015 [9]. The patients were included by 81 cardiologists from the French Region of Nord-Pas-de-Calais during outpatient visits. Documented CAD was defined as a history of myocardial infarction (MI), coronary revascularization, and/or the presence of coronary stenosis >50% on a coronary angiogram. Documented AF was defined as a history of AF, even if in sinus rhythm at inclusion. The sole exclusion criterion was age < 18 years. Patients with other cardiovascular or non-cardiovascular illnesses or co-morbidities were not excluded.

A case record form was completed at the initial visit with information regarding demographic and clinical details of the patients, including current medications. The treating cardiologists then followed up with the patients, with the number of outpatient visits at clinician discretion. Protocol-specified follow-up was performed at three years using a standardized case record form to report clinical events. In the case of missing information, a research technician contacted general practitioners and/or patients. The identification of patients with events for adjudication was based on interviews with patients/relatives during outpatient visits, discharge summaries for hospitalization during follow-up that were sent to treating cardiologists, and information obtained by the research technician. The events that patients reported were systematically confirmed from the medical reports.

This study was approved by the French medical data protection committee and authorized by the Commission Nationale de l’Informatique et des Libertés for the treatment of personal health data. All patients consented to the study after being informed in writing of the study’s objectives and treatment of the data, as well as about their rights to object and about access and rectification.

### 2.2. Study Design and Definitions

Figure 1 shows the study flow chart. Among the 10517 outpatients included in the CARDIONOR registry, a total of 6313 had documented CAD. We excluded 1282 patients with prevalent AF, leaving 5031 CAD patients with no history of AF at registry inclusion. For the present analysis, we focused on the 5015 patients (99%) for whom follow-up was available. Two investigators adjudicated incident AF, with a third opinion sought in cases of disagreement.

The diagnostic circumstances of AF, as well as the antithrombotic strategy, were systematically assessed and adjudicated. No specific screening was performed for AF detection; documented AF episodes, therefore, represented daily practice. Data on therapeutic management represent the initial cardiologist recommendation, as described in the medical report associated with the AF diagnosis. Patients with implanted devices who had documented atrial high rate episodes and whose treating cardiologists had diagnosed them as having probable AF (as documented in the medical report) were adjudicated as incident AF. HF was defined as a history of hospitalization for HF and/or a history of symptoms and signs of HF associated with echocardiographic evidence of systolic dysfunction, left ventricular hypertrophy, left atrial enlargement, or diastolic dysfunction. Cause of death was determined after a detailed review of the circumstances of death and classified as cardiovascular or non-cardiovascular, as previously defined [10]. Death by an unknown cause was kept as a separate category.

### 2.3. Statistical Analysis

Continuous variables are described as mean ± standard deviation (SD). Categorical variables are presented as absolute numbers and/or percentages. The incidence of AF was estimated with the cumulative incidence function, with death as the competing event. Univariable and multivariable assessments of baseline variables associated with incident AF were performed with the use of a cause-specific hazard model [11,12]. Hazard ratios (HRs) and 95% confidence intervals (CIs) were calculated. The proportional hazards assumption was tested visually using Kaplan–Meier curves and by examining plots of −ln [−ln (survival time)] against the ln (time). For continuous variables, the linearity assumption was assessed by plotting Schoenfeld residuals versus time. Collinearity was excluded by constructing a correlation matrix between candidate predictors. The comparison of baseline variables in patients with incident AF according to antithrombotic treatment was performed using the χ^2^ test, the Fisher’s exact test for categorical variables, and the Student’s unpaired t test for continuous variables. The associations between incident AF and mortality were assessed with Cox analyses, and incident AF was modeled as a time-dependent variable. HRs and 95% CIs were calculated. All statistical analyses were performed using STATA 14.2 software (STATA Corporation, College Station, TX, USA). Significance was assumed at *p* < 0.05.

## 3. Results

### 3.1. Study Population

A clinical follow-up was obtained at a median of 3.3 (interquartile range: 3.0 to 3.6) years in 5015 (99%) of the 5031 CAD outpatients without prevalent AF. As shown in Table 1, most patients were male (77.8%), with a mean age of 66.1 ± 11.7 years. A history of MI was documented in 50.7% of the cases, with 72.7% of the patients having had previous percutaneous coronary intervention (PCI) and 19.5% with a previous coronary bypass (CABG). The mean left ventricular ejection fraction (LVEF) was 57 ± 11%, and 18.2% of the patients had LVEF <50%. Secondary prevention medications were widely prescribed (antiplatelet agents 98%, statins 92.5%, angiotensin-converting enzyme inhibitors or angiotensin receptor blockers 83.2%, beta-blockers 82.4%).

### 3.2. Incident AF

During the follow-up period, there were 495 deaths (cardiovascular deaths: *n* = 200) among the 5015 patients. During the same period, 266 patients experienced real-life incident AF. Risk of AF increased progressively, with cumulative incidences including death as the competing event of 1.6% (95% CI: 1.3 to 1.9), 2.9% (95% CI: 2.4 to 3.3), and 4.7% (95% CI: 4.1 to 5.3) at years 1–3, respectively. Figure 2A shows the cumulative incidence of AF over the time and Figure 2B according to age at inclusion.

We performed univariable and multivariable assessments of baseline variables that might be associated with incident AF (Table 1 and Table 2). Five variables determined at registry inclusion were independently associated with incident AF: older age (*p* < 0.0001), heart failure (*p* = 0.003), lower LVEF (*p* = 0.008), history of hypertension (*p* = 0.010), and diabetes mellitus (*p* = 0.033). Of note, a history of MI was not associated with an increased risk for incident AF.

### 3.3. Diagnosis and Management of Incident AF

As shown in Figure 3A, the diagnosis of AF in the 266 patients took place in different settings. In two thirds of cases, incident AF was diagnosed during cardiology outpatient visits. Almost half of the patients in these situations had no evident symptoms of AF. Other relatively frequent diagnostic circumstances included hospitalization for heart failure (*n* = 46) and monitoring of implanted devices (*n* = 15). Of note, the number of patients who had AF diagnosed at the time of hospitalization for ischemic stroke was low (*n* = 5).

We assessed the antithrombotic strategy that was chosen in patients with incident AF. The mean CHA_2_DS_2_-VASc score in the 266 patients was 4.3 (±1.5). The proportion of women with a CHA_2_DS_2_-VASc score ≥ 3 was 97%, and the proportion of men with a CHA_2_DS_2_-VASc score ≥ 2 was 96%. As shown in Figure 3B, most patients were prescribed an anticoagulant (any anticoagulant: *n* = 245 (92%); direct oral anticoagulant: *n* = 127; vitamin K antagonist: *n* = 110; low-molecular-weight heparin: *n* = 8). When anticoagulation was not used, 12 patients received single-antiplatelet therapy and 8 patients received dual-antiplatelet therapy; one patient had no antithrombotic therapy. When an anticoagulant was used, the antithrombotic regimen also included an antiplatelet drug in half of cases (anticoagulant alone: *n* = 123; anticoagulant + single-antiplatelet therapy: *n* = 111; anticoagulant + dual-antiplatelet therapy: *n* = 11). At time of incident AF, 26 of the 266 patients had a recent (<1 year) history of MI and/or PCI. When focusing on the 240 remaining patients who experienced incident AF in the context of chronic CAD (i.e., previous MI and/or PCI > 12 months) (Figure 3C), an anticoagulant was used in 225 (94%), still often combined with an antiplatelet drug (anticoagulant alone: *n* = 121; anticoagulant + single-antiplatelet therapy: *n* = 102; anticoagulant + dual-antiplatelet therapy: *n* = 2). Apart from higher proportions of previous PCI (75% vs. 54.6%, *p* = 0.001) and previous stroke (8.7% vs. 1.7%, *p* = 0.026), patients who received anticoagulant and antiplatelet therapy had similar characteristics to patients treated with anticoagulant alone (Table 3).

### 3.4. Outcome After Incident AF

For the 266 patients with incident AF, the median clinical follow-up after AF diagnosis was 1.2 (interquartile range: 0.5 to 2.2) years. A total of 42 deaths (cardiovascular deaths: *n* = 26) occurred during the post-AF period. Table 4 shows the impact of incident AF, analyzed as a time-dependent variable, on all-cause and cardiovascular mortality. In adjusted models, incident AF during follow-up was associated with significant increases in the risk of all-cause mortality (HR: 2.04; 95% CI: 1.47 to 2.83) and cardiovascular mortality (HR: 2.88; 95% CI: 1.88 to 4.43).

## 4. Discussion

Interest is growing in analyzing outcomes in patients with chronic CAD [13,14,15]. Incident MI [16] or incident stroke [17] are probably the first events clinicians think of when assessing risk in these patients. However, other cardiovascular events may also affect management and may have significant prognostic implications. Our study documents a relatively high risk of real-life incident AF in chronic CAD patients, with a roughly linear increase of 1.6% per year. This result should be interpreted in the context of an unselected population of consecutive chronic CAD outpatients with a significant proportion of elderly individuals, and frequent history of hypertension, diabetes mellitus, and heart failure, all factors that are associated with incident AF in the present study as well as in the literature [7,18,19]. Incident AF is much less important in general population as reported by Vermond et al. in a large Dutch cohort with a cumulative incidence of 3% after a ten years mean follow-up and 3.3/1000 person-years [7]. In line with this, Wike et al. reported in a larger German cohort an incidence of 4.112/1000 person-years in the general population [20]. Importantly, to the best of our knowledge, no study assessed the incidence of AF in CAD outpatients. Of note, the interpretation of the rate of incident AF implies a need to consider screening strategies in this population. Our study protocol did not require specific screening for AF, so our data document incident AF diagnosed during routine real-life follow-up. Also, from a methodological point of view, we emphasize that our study may differ from earlier literature in that we present cumulative incidences, taking into account death as the competing event. We justify this choice by the high mortality rate of CAD patients at risk for AF. Indeed, inappropriate censoring of competing events may lead the Kaplan–Meier estimator to overestimate the cumulative incidence in the presence of competing risks, especially if the competing risk is frequent [11,12].

As stated above, the real-life design of our study yielded information on the diagnostic circumstances of incident AF. Such data are rarely available in the literature. An integral part of management for patients with chronic CAD is the planning of regular follow-up visits with the cardiologists. Although chronic CAD guidelines recommend an annual resting electrocardiogram (ECG), the level of evidence is acknowledged to be low [4,5]. Our data showed that incident AF was frequently diagnosed by a systematic ECG in the absence of AF-related symptoms. This high proportion of asymptomatic patients could be related to the wide use of betablockers prior to AF occurrence in our population. Moreover, it is plausible (although speculative) that patients with a history of CAD who experienced new symptoms had facilitated access to cardiology advice. This may have had important consequences by minimizing treatment delays. Concordant with these data is the relatively low number of incident AF discovered at time of hospitalization for an ischemic stroke (*n* = 5 for 5015 CAD patients followed-up during three years; 0.3/1000 patient-years).

International taskforces currently recommend against systematic screening for AF in the general population, citing the cost implications and uncertainty over the benefits of a systematic screening program compared to usual care [21]. However, screening in targeted high-risk groups remains to be questioned. Given the high incidence of AF compared to the general population and the proportion of asymptomatic AF in CAD outpatients reported in our study, extended screening strategies in such patients would be of interest.

One aim of our analysis was to describe the management strategy when incident AF is detected in chronic CAD patients. The present study focused on initial management at the time of AF diagnosis and, as such, clearly differs from previous analyses of registries reporting chronic medications in patients combining CAD and AF [3,22,23]. In addition, the 266 cases of incident AF occurred between 2013 and 2018, so our study describes the modern management of incident AF in patients with a history of CAD. First, we documented a very high use of anticoagulation, which is in accordance with the high thrombotic risk of the study population as documented by the CHA_2_DS_2_-VASc score. Indeed, according to current guidelines [24], anticoagulation would have to be considered in almost all CAD patients experiencing incident AF in the present study. Second, because almost all patients had a background of antiplatelet therapy before the AF event, the management of combinations of antithrombotic drugs was a matter of interest. When focusing the analysis on chronic patients, we found that cardiologists are still reluctant to stop all antiplatelet therapy in these patients. AF guidelines suggest going with anticoagulation alone if >1 year has passed with no acute events [24]. However, the level of evidence is limited, and expert consensus provides more modulated recommendations [25]. One recent randomized trial and many observational studies have shown that the addition of antiplatelet therapy is associated with a substantially increased risk of bleeding, with no clear benefit on ischemic events [3,22,23,26].

Finally, incident AF has been associated with increased mortality in general populations [7,8]. Our study extends these findings to a large population of outpatients with chronic CAD. In adjusted analyses, CAD patients who developed AF had a two-fold increased risk for all-cause mortality, largely similar to associations previously reported [7,8]. Incident AF should, therefore, be considered as an important warning sign for physicians working with CAD patients, even if anticoagulation is largely used. These data are concordant with previous findings suggesting that next to stroke prevention, further research is needed to improve the prognosis of patients with AF [7,27,28].

### Study Limitations

Our study has some limitations. First, our data reflect the practice in a regional area, and we do not know whether these findings are generalizable for practices in other parts of the world. Second, because cardiologists determined inclusion, the data may not be generalizable to the overall population with CAD in the community. Finally, we present here initial management strategies for patients with incident AF and lack details on chronic management and antithrombotic modifications during follow-up. On the other hand, the absence of exclusion criteria, the very high follow-up rate, and the adjudication of clinical events can be considered strengths of the study.

## 5. Conclusions

Our study shows that real-life incident AF occurs at a stable annual rate of 1.6% in chronic CAD outpatients. Older age, heart failure, low LVEF, hypertension, and diabetes were associated with a higher risk of AF. In patients with chronic CAD, a substantial proportion of incident AF is diagnosed during a systematic cardiology outpatient visit in asymptomatic patients. In patients with chronic CAD and incident AF that were >1 year from their last MI and/or PCI, antiplatelets remain frequently combined with oral anticoagulation. Finally, we found that incident AF in patients with chronic CAD was associated with an increase in all-cause and cardiovascular mortality. Considering the high incidence of AF compared to the general population and the proportion of asymptomatic AF in CAD outpatients, extended screening strategies in such patients would be of interest.

## Figures and Tables

**Figure 1 jcm-09-02367-f001:**
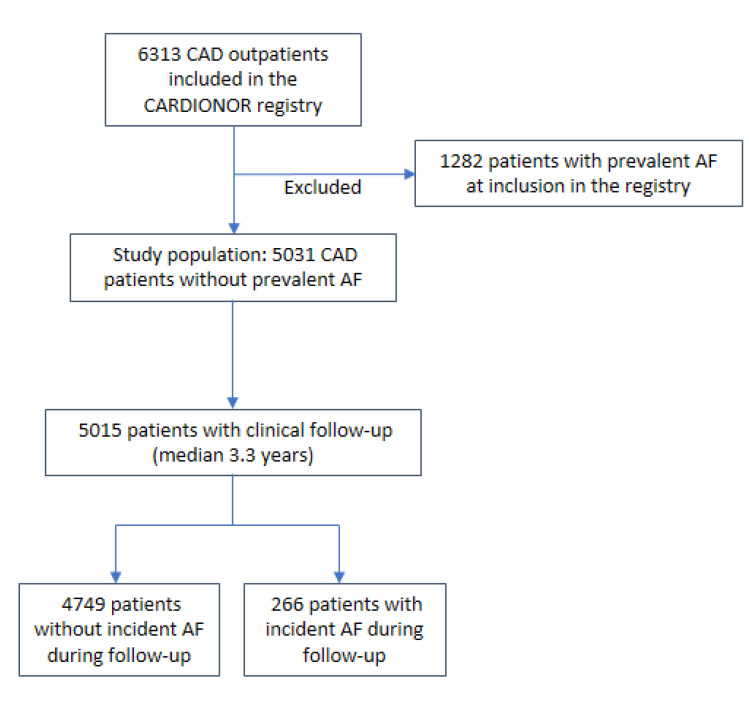
Study flow chart.

**Figure 2 jcm-09-02367-f002:**
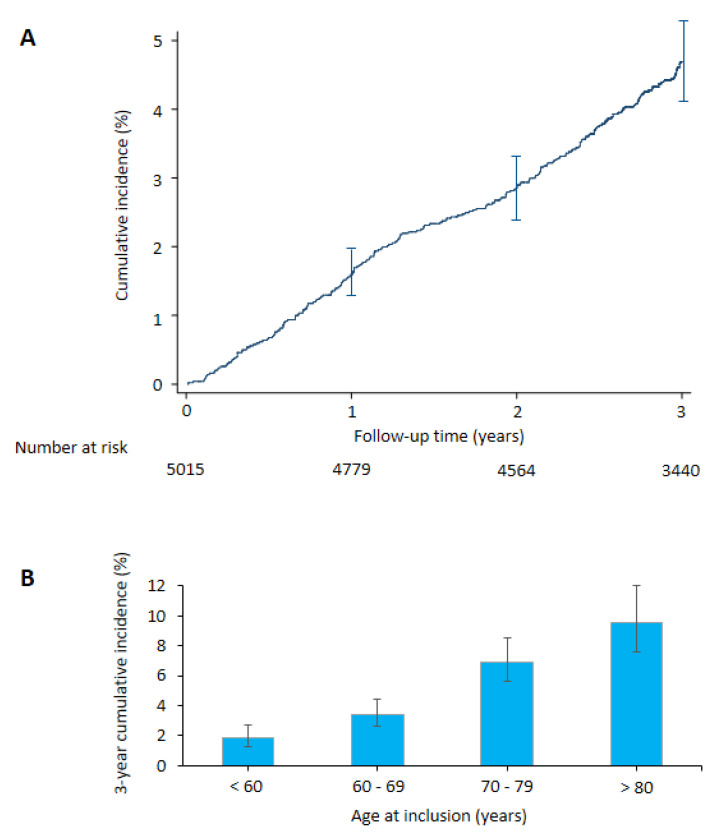
Incidence of a first episode of atrial fibrillation (AF). (**A**) Cumulative incidence of AF during the follow-up period (death as the competing event). (**B**) 3-year cumulative incidence of AF (death as the competing event) according to age at inclusion. Error bars are 95% CI.

**Figure 3 jcm-09-02367-f003:**
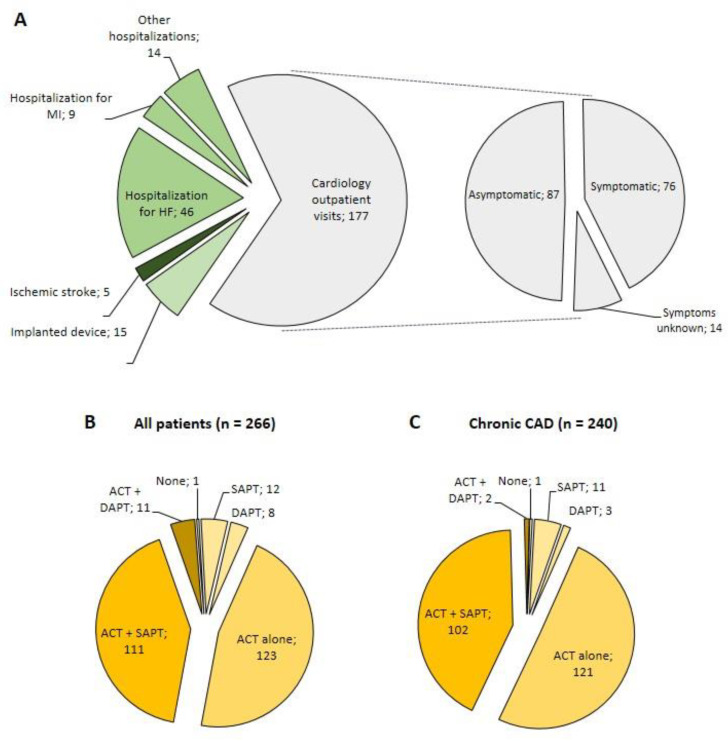
Diagnostic circumstances and antithrombotic management of incident atrial fibrillation (AF) in coronary artery disease (CAD) outpatients. (**A**) Diagnostic circumstances of incident atrial fibrillation (AF) in coronary artery disease (CAD) outpatients. HF, heart failure; MI, myocardial infarction (**B**) Antithrombotic strategy in all coronary artery disease (CAD) outpatients with incident atrial fibrillation (AF); ACT, anticoagulant therapy; DAPT, dual-antiplatelet therapy; SAPT, single-antiplatelet therapy. (**C**) Antithrombotic strategy in patients with incident AF in a context of chronic CAD (i.e., patients without recent (<1 year) history of myocardial infarction and/or percutaneous coronary intervention).

**Table 1 jcm-09-02367-t001:** Baseline characteristics of the study population and correlates of incident atrial fibrillation (AF) according to univariable analysis.

	All Patients with Follow-Up (*n* = 5015)	No Incident AF (*n* = 4749)	Incident AF (*n* = 266)	HR [95% CI]	*p*
Age, years	66.1 ± 11.7	65.8 ± 11.6	72.6 ± 10.4	1.06 [1.05–1.07]	<0.0001
Women	22.2	21.9	27.8	1.37 [1.05–1.79]	0.021
History of hypertension	59.2	58.4	73.2	1.95 [1.48–2.56]	<0.0001
History of diabetes mellitus	31.7	31.3	39.9	1.46 [1.14–1.86]	0.003
Previous MI	50.7	50.8	49.2	0.96 [0.75–1.22]	0.715
Previous PCI	72.7	73.1	65.8	0.69 [0.53–0.88]	0.004
Previous coronary bypass	19.5	19.3	22.9	1.25 [0.94–1.66]	0.127
Previous stroke	4.7	4.6	6.0	1.39 [0.84–2.30]	0.202
History of peripheral artery disease	23.5	23.3	27.1	1.25 [0.96–1.64]	0.104
Heart failure	14.5	13.8	27.8	2.67 [2.04–3.50]	<0.0001
LVEF, %	57 ± 11	57 ± 10	54 ± 13	0.97 [0.96–0.98]	<0.0001
LVEF < 50%	18.2	17.6	28.2	1.95 [1.49–2.54]	<0.0001
Medications at inclusion:					
Antiplatelet drug	98.0	97.9	98.9	1.80 [0.58–5.62]	0.311
Oral anticoagulant	4.1	4.1	3.4	0.81 [0.42–1.57]	0.533
At least 1 antithrombotic drug	99.3	99.3	99.6	1.82 [0.26–13.0]	0.550
Angiotensin-Converting enzyme inhibitor or angiotensin receptor blocker	83.2	82.9	88.4	1.53 [1.05–2.22]	0.027
Beta-Blocker	82.4	82.2	86.8	1.39 [0.98–1.99]	0.067
Statin	92.5	92.7	90.2	0.72 [0.48–1.07]	0.107

Data are presented as mean ± standard deviation (SD) or %. HR, hazard ratio; CI, confidence interval; MI, myocardial infarction; PCI, percutaneous coronary intervention; LVEF, left ventricular ejection fraction.

**Table 2 jcm-09-02367-t002:** Independent correlates of incident atrial fibrillation (AF) by multivariable analysis.

	HR [95% CI]	*p*
Age (per year)	1.05 [1.04–1.07]	<0.0001
Heart failure	1.67 [1.19–2.35]	0.003
LVEF (per %)	0.98 [0.97–0.99]	0.008
History of hypertension	1.45 [1.09–1.93]	0.010
History of diabetes mellitus	1.31 [1.02–1.69]	0.033

HR, hazard ratio; CI, confidence interval; LVEF, left ventricular ejection fraction. The variables included in the model were age, sex, history of hypertension, history of diabetes mellitus, previous myocardial infarction, previous percutaneous coronary intervention, previous coronary bypass, previous stroke, history of peripheral artery disease, heart failure, and LVEF. A stepwise approach was used with forward selection (the *p* value for entering into the stepwise model was set at 0.05).

**Table 3 jcm-09-02367-t003:** Comparison of patients receiving anticoagulant therapy (ACT) alone vs. ACT and antiplatelet therapy (APT) (*n* = 225 patients with incident atrial fibrillation (AF) and without a recent (<1 year) history of myocardial infarction (MI) or percutaneous coronary intervention (PCI)).

	ACT Alone (*n* = 121)	ACT + APT (*n* = 104)	*p*
Baseline characteristics			
Age, years	73.5 ± 10.0	71.5 ± 10.8	0.137
Women	29.8	25.0	0.426
History of hypertension	69.4	74.8	0.376
History of diabetes mellitus	37.2	44.2	0.283
Previous MI	46.3	52.9	0.323
Previous PCI	54.6	75.0	0.001
Previous coronary bypass	30.6	20.2	0.076
Previous stroke	1.7	8.7	0.026
History of peripheral artery disease	28.9	22.1	0.244
Heart failure	24.8	32.7	0.190
LVEF, %	55 ± 12	53 ± 14	0.149
AF diagnosis			
Cardiology outpatient—asymptomatic	37.2	32.7	0.481
Cardiology outpatient—symptomatic	30.6	28.9	0.777
Hospitalization for heart failure	16.5	19.2	0.597
Implanted device	7.4	3.9	0.391
CHA_2_DS_2_-VASc score at AF diagnosis	4.3 ± 1.5	4.3 ± 1.3	0.700

Data are presented as mean ± SD or %. LVEF, left ventricular ejection fraction.

**Table 4 jcm-09-02367-t004:** Association of incident atrial fibrillation (AF) with mortality.

	HR [95% CI]	*p*
**All-Cause Mortality**		
unadjusted	3.90 [2.82–5.37]	<0.0001
adjusted	2.04 [1.47–2.83]	<0.0001
**Cardiovascular Mortality**		
unadjusted	6.49 [4.26–9.89]	<0.0001
adjusted	2.88 [1.88–4.43]	<0.0001

HR, hazard ratio; CI, confidence interval; incident AF was used as a time-dependent variable. Adjusted models included age, sex, history of hypertension, history of diabetes mellitus, heart failure, and LVEF.

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
