# Peer review of "Real-Life Incident Atrial Fibrillation in Outpatients with Coronary Artery Disease"

_jcm, 2020, doi:10.3390/jcm9082367_

Round 1

Reviewer 1 Report

The authors of this manuscript present the results of the incidence of atrial fibrillation during a 3-year follow-up in coronary artery disease outpatients included in the prospective multicenter CARDIONOR registry.

However, I wonder if the manuscript I have access to is the correct form, as I don’t see the number of pages and lines. I can’t find neither 'Supplementary material? at Page 2 at paragraph 2.2. Study design and definitions. Authors mention ‘The flow chart of the present study is provided in supplementary material’.

One of the risk factors were heart failure and LVEF, which criteria did the authors consider to define having heart failure (by clinical report?), I would suggest to clarify

Table 3, is split in two pages 

Figure 2, should be before table 3 (according to the text)

Abbreviations, some are missing (i.e APT, DAPT, SAPT), I would recommend checking and add those missing

Author Response

We thank the editors and reviewers for their interest in our study and very constructive comments. As requested, we modified our previous manuscript with a rewrite based on the points raised by the 2 reviewers. As suggested by the editors, we are now submitting our revised manuscript.

Reviewer #1: 

The authors of this manuscript present the results of the incidence of atrial fibrillation during a 3-year follow-up in coronary artery disease outpatients included in the prospective multicenter CARDIONOR registry.

  1. However, I wonder if the manuscript I have access to is the correct form, as I don’t see the number of pages and lines. I can’t find neither 'Supplementary material? at Page 2 at paragraph 2.2. Study design and definitions. Authors mention ‘The flow chart of the present study is provided in supplementary material’.

Authors: We thank the reviewer for this comment. We apologize for this mistake. To fix this point, pages and lines numbers have been added in the present manuscript. Furthermore, the flow chart has been added as supplementary materials.

  1. One of the risk factors were heart failure and LVEF, which criteria did the authors consider to define having heart failure (by clinical report?), I would suggest to clarify.

Authors: We thank the reviewer for this important point. The methods of the Cardionor study have been previously published (1). To address this point the definition of heart failure is now further described in the methods: line 85: “HF was defined as a history of hospitalization for HF and/or a history of symptoms and signs of HF associated with echocardiographic evidence of systolic dysfunction, left ventricular hypertrophy, left atrial enlargement or diastolic dysfunction.”

(1) Lamblin N, Ninni S, Tricot O, et al. Secondary prevention and outcomes in outpatients with coronary artery disease, atrial fibrillation or heart failure: a focus on disease overlap. Open Heart 2020; 7: e001165. DOI: 10.1136/openhrt-2019-001165.

  1. Table 3, is split in two pages

Authors: We apologize for this format mistake. Now the table 3 appears entirely at page 8.

  1. Figure 2, should be before table 3 (according to the text)

Authors: We thank the reviewer for this remark. As requested, the Figure 2 now appears prior to table 3

Reviewer 2 Report

Ninni and colleagues submitted a nice and complete analysis of the Atrial Fibrillation incidence among patients with Coronary Artery Disease. The strong study design, based on reliable and sufficient data provides convincing conclusions. However, few additional informations are needed to help the reader acquire the importance of this work, from a broader perspective.

1/ Could the authors compare the 1.6% incidence rate to the average one found in the general population based on previous epidemiological studies or publications ?

2/ Could they also compare the AF incidence in CAD patients to AF incidence the general population?

3/ Could they compare the AF incidence to the population without CAD ?

4/ Could the authors emphasize the novelty of their key result within their manuscript, and highlight its clinical implication ?

5/ Unless requested, could the authors correct the police size of the last listed authors ?

Author Response

We thank the editors and reviewers for their interest in our study and very constructive comments. As requested, we modified our previous manuscript with a rewrite based on the points raised by the 2 reviewers. As suggested by the editors, we are now submitting our revised manuscript.

Reviewer #2:

  1. Could the authors compare the 1.6% incidence rate to the average one found in the general population based on previous epidemiological studies or publications? Could they also compare the AF incidence in CAD patients to AF incidence the general population?

Authors: We thank the reviewer for this important comment. Indeed AF incidence in CAD outpatient is much higher than general population. In a publication from Vermond et al. investigating AF incidence in a large Dutch cohort, a cumulative incidence of 3% was reported after a 9.7 years follow-up (3.3/1000 persons-years). Here we report a cumulative incidence of 4.7% at 3 years.

This point is now further discussed line 237: “Incident AF is much less important in general population as reported by Vermond et al. in a large Dutch cohort with a cumulative incidence of 3% after a ten years mean follow-up and 3.3/1000 person-years. In line, Wike et al reported in a larger German cohort a 4.112/1000 person-years in general population”

  1. Could they compare the AF incidence to the population without CAD ?

Authors: We thank the reviewer for this point. To the best of our knowledge, no study assessed so far the incidence of AF according to CAD status or focus on non-CAD patients. To highlight this point, we have added a sentence line 240: “Importantly, to the best of our knowledge, no study assessed the incidence of AF in CAD outpatients.”

  1. Could the authors emphasize the novelty of their key result within their manuscript, and highlight its clinical implication?

Authors: We thank the reviewer for this important comment.

To highlight the impact of our findings, we have added a sentence in the conclusion paragraph line 306: “Considering the high incidence of AF compared to general population and the proportion of asymptomatic AF in CAD outpatients, extended screening strategies in such patients would be of interest.”

  1. Unless requested, could the authors correct the police size of the last listed authors?

Authors: We apologize for this mistake. Police size of the last listed authors has been resized correctly.

Round 2

Reviewer 1 Report

The authors have put a considerable amount of effort in revising this manuscript according to the reviewer's instructions.

Couple of issues are still problematic:

. Major Issue, I would suggest the authors to place the flow chart at the main document

. Minor issues. Figure 1 at page 5 is repeated at page 6. Figure 2 at page 8, is repeated at page 10 (I understand are to delete after review)

Author Response

We thank the editors and reviewers for their interest in our study. As requested, we modified our previous manuscript with a rewrite based on the points raised by the reviewer. As suggested by the editors, we are now submitting our revised manuscript.

Reviewer #1: 

The authors have put a considerable amount of effort in revising this manuscript according to the reviewer's instructions.

Couple of issues are still problematic:

  • Major Issue, I would suggest the authors to place the flow chart at the main document

Authors: We thank the reviewer for this important comment. We recognize the importance of providing the study flow chart in the main document. Therefore, the flow chart appears as figure 1 in the methods section.

  • Minor issues. Figure 1 at page 5 is repeated at page 6. Figure 2 at page 8, is repeated at page 10 (I understand are to delete after review)

Authors: Indeed, figure 1 and 2 appear twice because of the tracked changes in the Word file. These doubled figured will disappear in the final version.